# TRPing to the Point of Clarity: Understanding the Function of the Complex TRPV4 Ion Channel

**DOI:** 10.3390/cells10010165

**Published:** 2021-01-15

**Authors:** Trine L. Toft-Bertelsen, Nanna MacAulay

**Affiliations:** Department of Neuroscience, University of Copenhagen, Blegdamsvej 3, 2200 Copenhagen N, Denmark; macaulay@sund.ku.dk

**Keywords:** TRP channels, transient receptor potential vanilloid 4, TRPV4

## Abstract

The transient receptor potential vanilloid 4 channel (TRPV4) belongs to the mammalian TRP superfamily of cation channels. TRPV4 is ubiquitously expressed, activated by a disparate array of stimuli, interacts with a multitude of proteins, and is modulated by a range of post-translational modifications, the majority of which we are only just beginning to understand. Not surprisingly, a great number of physiological roles have emerged for TRPV4, as have various disease states that are attributable to the absence, or abnormal functioning, of this ion channel. This review will highlight structural features of TRPV4, endogenous and exogenous activators of the channel, and discuss the reported roles of TRPV4 in health and disease.

## 1. Introduction

Transient receptor potential (TRP) channels can be considered as multiple signal integrators directing our sensory systems. The TRP channels, however, possess a broader role than classical sensory transduction, as they respond to all manner of stimuli both from within and from outside the cell. Members of the TRP superfamily—which is diverse and encompasses 28 TRP channel genes—share the common features of six transmembrane segments with N- and C-termini residing in the cytoplasm, the former of which contains at least three ankyrin repeats, in addition to their permeability to cations [1,2,3,4].

TRP channels are grouped into six major subfamilies based on nucleotide sequence homology: TRPM (melastin); TRPC (canonical); TRPV (vanilloid); TRPP (polycystin); TRPML (mucolipin); and TRPA (ankyrin). The physiological role of these TRP channels has become apparent by the wide array of human diseases that are now attributable to dysfunction of these channels, for review see [5,6]. The vanilloid subfamily of TRP channels can be further subdivided into six isoforms, of which the fourth member, TRPV4, has emerged as a regulator of cell function [7] in multiple tissues since being discovered in 2000 [8,9] as a homologue of the invertebrate *Caenorhabditis elegans* gene Osm-9. TRPV4 is characterized by multimodal activation properties that implicate it in a broad range of functions [10]. However, the mechanisms underlying activation of the channel by differing modalities remain largely elusive. 

Here, we review the characteristics of this ion channel’s structure, localization, function, and regulation prior to discussing its functional importance in health and disease.

### 1.1. TRPV4 Expression, Structure, and Isoforms

Expression of TRPV4 in mammals is wide-spread with membranous expression detected in the brain [11,12,13,14], the eyes [15,16,17,18,19], kidney tissue and the urinary system [20,21,22,23,24], the gastrointestinal tract and pancreas [25,26], the skin [27], in several musculoskeletal tissues [28,29,30], in epithelia [20,31,32,33], and in vasculature (both endothelium [34,35,36,37,38] and surrounding smooth muscle cells [13,38,39,40,41,42,43,44]) (Figure 1).

The *trpv4* gene belongs to both the TRP (transient receptor potential) cation channel and the ANKRD (ankyrin repeat domain) containing family of genes. It is found on chromosome 12 at position 24.1 (https://ghr.nlm.nih.gov/gene/TRPV4) and is composed of 15 exons. TRPV4 exists as five different splice variants (TRPV4A-E) (Figure 2) that can be grouped into two classes: Group I (TRPV4-A and TRPV4-D) and Group II (TRPV4-B, TRPV4-C, and TRPV4-E). Group I contains the TRPV4 variant encoded by the canonical sequence, which is known as TRPV4-A, and the TRPV4-D, which lacks residues 27–61. Both variants localize to the cell membrane and display identical channel activity and regulatory properties [45]. The Group II splice variants (B, C, E) contain N-terminal deletions, which cause impaired oligomerization and subsequent intracellular retention. These channel products are thus detected in the endoplasmatic reticulum (ER) [45]. The TRPV4-A (henceforth referred to as TRPV4) contains 871 amino acids and exhibits a topology consisting of six predicted transmembrane domains (TM) with linking loops and a putative re-entrant pore loop between TM5 and TM6, and intracellular N- and C-termini, which face the cytoplasm and contain a variety of functional domains [6]; a topology that resembles that of voltage-gated ion channels. Although the crystal structure of the channel awaits, insight into its three-dimensional structure has been obtained by cryo-electron microscopy with a resolution of 3.5 nm [46]. The ankyrin repeat domains (ARD; at least three domains) detected in the N-terminus form a double-helical structure connected by extraordinarily long folds, which might be important for protein–protein interactions [47,48,49]. In addition, the N-terminus presents with a proline-rich domain (PRD) [46] and lies on the membrane in an oblique orientation [46]. The TRPV4 C-terminal tail contains additional functional domains such as a TRP box, a calmodulin-binding site, and an amino acid stretch (Asp-Ala-Pro-Leu) similar to a PDZ-binding like motif (http://www.trpchannel.org).

The biogenesis of TRPV4 involves glycosylation (N-glycosylation on residue N651 (3)) and oligomerization in the ER with a final transfer to the Golgi apparatus for subsequent maturation. The channel usually organizes in a homotetrameric structure [45], in a manner requiring both the N- and the C-termini [16,17], while heterotetramers may occur with TRPC1 [50], TRPP2 [51], and TRPC1-TRPP2 [52].

### 1.2. Biophysical Properties and Ionic Permeability

The TRPV4 ion channel is a non-selective cation channel (Ca^2+^, Mg^2+^ or Na^+^ as the permeating cations) and is characterized by a moderately high Ca^2+^ permeability ratio (PCa^2+^/PNa^+^ = 6–10, PMg^2+^/PNa^+^ = 2–3 [2,53]). The ionic pathway is generated by a pore-forming loop between the S5 and S6 domain (Figure 3), in which the residues Asp672 and Asp682 modulate the ion selectivity: alanine substitution of either or both of these asparagines causes a reduction in the relative permeability for divalent cations and of the degree of outward rectification [2]. Most studies obtain a linear current–voltage relationship when recording TRPV4-mediated currents in retinal Müller glia, microvascular endothelial cells (forming the inner retinal blood–retinal barrier) [54], non-pigmented epithelial cells [18], and TRPV4-expressing *Xenopus laevis* oocytes [55,56]. However, removal of Na^+^ and Ca^2+^ from the test solution promoted current–voltage relationships with outward rectification in TRPV4-expressing HEK293 cells and oocytes [1,2,57,58].

## 2. Gating of TRPV4

TRPV4 channels display remarkable gating promiscuity and are activated by a palette of stimuli such as endogenous ligands, cell swelling, temperature, and synthetic/exogenous ligands (see below). However, the mechanism(s) underlying TRPV4 activation by these varying modalities are currently poorly understood.

### 2.1. Endogenous Ligands

TRPV4 appears to be activated by endogenous compounds, specifically those that are products of enzymatic phospholipase A2 (PLA2) activity [59]. PLA2-mediated release of arachidonic acid and its epoxyeicosatrienoic acid metabolites (5′,6′-EET) directly activates TRPV4 in retinal Müller cells and TRPV4-expressing HEK293 cells [16,34]. The 5,6-EET-mediated modulation is proposed to take place via the newly discovered EET-binding pocket on the S2-S3 linker of TRPV4 [59] and/or indirectly by modulating the membrane fluidity in the vicinity of the ion channel [60]. In other cell types, i.e., retinal ganglion cells, yeast, choroid plexus epithelial cells, and TRPV4-expressing *Xenopus laevis* oocytes, TRPV4 activity remains undisturbed upon PLA2 activation and/or exposure to the downstream effectors [16,61,62]. In order to bypass PLA2 activation, these cell types were exposed to a range of enzymatic products and their metabolites, which all failed to activate TRPV4 in these cell types, whether applied extra- or intracellularly [16,62,63]. It thus remains unresolved how such metabolites can activate TRPV4 in certain cell types, but not in others.

Dimethylallyl pyrophosphate (DMAPP), an intermediate used for the cholesterol synthesis in the mevalonate metabolic pathway, has been identified as an endogenous agonist of sensory neuronal TRPs affecting receptor-specific nociception [64,65], thus being a useful tool to understand the molecular basis for TRP-related peripheral pain mechanisms. With the emerging range of exogenous ligands able to activate TRPV4, taken together with its basal activity in many cell types, one may expect that a growing list of endogenous activators awaits discovery.

### 2.2. Cell Swelling

Maintenance of cell volume is a homeostatic imperative for cells of most origins [66]. Because cellular structures in tissues and organs may frequently be exposed to challenges that cause mechanical stress and/or volume changes, cellular responses that are suitable for achieving an appropriate physiological reaction to the concerned stressor in question are set in motion. Initially, TRPV4 was described as an osmo-sensor, activated by hyposmolar stress [8,9,67]. The osmotically-induced activation of TPRV4 came about following experimental exposure to large hyposmotic gradients (in the order of 100–150 mOsm) [16,55,68] and was proposed to require an obligatory formation of macromolecular complexes with a specific aquaporin (AQP), such as AQP2 [69], AQP4 [56,68,70], or AQP5 [71]. However, it appeared that TRPV4 could be activated by osmotically-induced cell swelling, irrespective of the AQP isoform [55], as the co-expressed AQP was required solely to translate an abruptly introduced experimental osmotic challenge into the swift volume increase needed to activate TRPV4 [16,55,56,72]. In fact, TRPV4 is activated by cell swelling, irrespective of the origin of the cell swelling, even in the absence of an AQP. This point was demonstrated following TRPV4 activation upon cell swelling occurring in the absence of an osmotic gradient by the water-transporting Na^+^/K^+^/2Cl^-^ cotransporter [56,72]. TRPV4 is, therefore, considered a sensor of altered cell volume rather than of the osmotic gradient a given cell faces, and is thus “volume-sensitive”, rather than ”osmo-sensitive” [55].

The molecular link coupling the cell swelling to TRPV4 activation remains a topic of controversy, the solution of which may reside in cell type-specific responses. Swelling-induced TRPV4 activation in Müller glia relies on an intermediate pathway involving the enzymatic activity of PLA2 and its downstream effectors [16,63]. In other cell types, i.e., retinal ganglion cells, yeast, and TRPV4-expressing *Xenopus laevis* oocytes, TRPV4 is readily activated directly by cell swelling in the absence of PLA2 activity (and its downstream metabolites) [16,62,63,73].

The TRPV4 N-terminus dictates the volume sensitivity of the ion channel, as replacement of the N-terminus with that of the volume-insensitive TRPV1 isoform produced a TRPV4 chimera with no volume sensitivity [55]. Replacement of the TRPV4 N-terminus with the N-terminus from the shrinkage-sensitive splice variant of TRPV1 (VR.5′sv) provided the designed TRPV4 chimera with the ability to be activated by cell shrinkage [63,74,75,76]. Stepwise deletions revealed that the distal-most part of the N-terminus, more specifically the proline-rich domain, was required for the response to cellular swelling [63]. This finding aligns with the cytoskeletal protein PACSIN3 interacting specifically with this section of the TRPV4 N-terminus [77], which thus may be the point of contact in the proposed interaction between TRPV4 and the cytoskeleton [78,79,80,81] underlying volume-induced activation of TRPV4. The TRPV4 N-terminus may, in addition, modulate the volume sensitivity of the ion channel by binding to the membraneous phosphatidylinositol-4,5-diphosphate, which leads to rearrangements of the tail region and Syndapin3/PACSIN3 [82].

### 2.3. Temperature

Maintenance of body temperature is a basic physiological process for many different organisms. TRPV4 is essential for basal physiological sensing of ambient temperature, via its expression in primary sensory neurons, skin keratinocytes, and the preoptic/anterior hypothalamus [9,58,83,84,85]. As with many members of the TRP channel superfamily, TRPV4 is gated by temperature changes [3,9,27,58,86]. The dynamic temperature range for TRPV4 generally falls within 27–44 °C, but fluctuates with cell type: TRPV4-mediated current responses occurred in TRPV4-expressing *Xenopus laevis* oocytes at temperatures around 27 °C [9], in HEK293 cells around 34 °C [3,58], in keratinocyte cells around 33°C [27], and in dissociated hippocampal pyramidal neurons at temperatures approaching physiological body temperature (>34 °C) [86]. Lastly, heat-mediated activation of TRPV4 has been proposed to require an endogenous signal, since heat activation of the channel failed to occur in inside-out patches of HEK293 cells [3].

Not only does temperature change alter the basal TRPV4-mediated current, it may also influence the volume-induced regulation of the channel [87], as observed with peak volume sensitivity of TRPV4-expressing CHO cells at physiological body temperature [9]. However, other reports failed to detect temperature-sensitivity of the volume-mediated TRPV4 activation [4,8], which could possibly reside in the non-physiologically high temperatures employed in these experiments (65 °C) [8] and/or prolonged suprathreshold temperature stimulus [51], which could cause desensitization of TRPV4, as previously described for the related TRPV1 [88]. Finally, disruption of unesterified membrane cholesterol content has been found to modulate the transduction of temperature stimulation of TRPV4 in mouse Müller cells [89].

### 2.4. Exogenous Ligands

During the experimental efforts to delineate the physiological role of TRPV4, a range of exogenous TRPV4 activators have been discovered [90], ranging from bisandrographolide A (BAA) originating from an extract of the Indian herbaceous plant *Andrographis paniculata*, citric acid, apigenin (4′,5,7-trihydroxyflavone, a flavone found in many plants), synthetic lipids, i.e., GSK1016790A and RN-1747, in addition to phorbol esters [91,92].

Phorbol esters, such as 4α-phorbol 12,13-didecanoate (4α-PDD), 4α-phorbol 12,13-dihexanoate (4α-PDH), and phorbol 12-myristate 13-acetate (PMA) have been demonstrated to induce membrane currents in TRPV4-expressing cells [35,93]. A direct activation of TRPV4 by these compounds [36,91] is thought to occur via a proposed binding site for the compounds between TM3 and TM4 [91] via interaction with specific amino acid side chains in TM4-6 (L584, T586, T591, R594 in TM4; F617L, Y621L, F624L in TM5; Y702L in TM6) [91,94]. TRPV4 activation by 4α-PDD may, however, occur by some indirect cellular regulatory pathway, as excessively long incubation times are required to obtain a stimulatory effect [36,91]. In addition, phorbol esters can activate membrane currents in neurons independently of TRPV4 [95], and their effect is more pronounced in ruptured cells than in their intact counterparts [35,36].

The synthetic lipid GSK1016790A is regularly employed as a molecular activator of TRPV4 [55,96,97]. GSK1016790A-mediated activation of TRPV4 occurs with an EC_50_ in the 3–30 nM range [18,96,97]. The specific interaction site is unknown, although the TRPV4-F54A mutant showed reduced response to the activator [40], and GSK1016790A-mediated activation of TRPV4 was later suggested to require the distal-most N-terminus of the channel [63]. A recent study demonstrated that GSK1016790A activated TRPV4 to an extent similar to that obtained by cell swelling, with no additive effect of the two TRPV4-activating regimes [55]. GSK10106790A thus appears to activate TRPV4 in a manner mimicking that of its physiologically-relevant and well-recognized swelling-induced gating event.

TRPV4 antagonists come as non-specific inhibitors with cross-reactivity with other TRP channels and stretch-activated channels; ruthenium red and gadolinium [2,98], but also as TRPV4-specific blockers such as RN-1734 [92, 96], RN-9893 [99], the anti-itch agent, crotamiton (N-ethyl-o-crotonotoluidide) [100], and HC-067047 [101], the latter of which demonstrated its potency in vivo by inhibiting rodent bladder hyperactivity with no obvious side effects [101].

## 3. Modulation of TRPV4

Approximately 70% of the TRPV4 channel’s total structure is located at the cytosolic side of the plasma membrane [46] and a range of protein–protein interactions and phosphorylations are proposed to take place in this region of TRPV4, leading to modification of channel trafficking, of the sensitivity to gating stimuli, or of the downstream signaling (see below).

### 3.1. Protein–Protein Interactions

TRPV4 may be retained in the ER by OS-9, which engages in protein–protein interaction with the TRPV4 N-terminus and thereby reduces the TRPV4 abundance at the plasma membrane [102]. STIM1, the stromal interaction molecule 1, is an auxiliary protein of TRPV4 channels, which is proposed to be required for trafficking of TRPV4 from the ER to the plasma membrane [103,104]. Endocytosis and membrane localization of TRPV4 is modulated by the cytoskeletal protein PACSIN 3 that binds to the proline-rich-domain of the TRPV4 N-terminus via the Src homology 3 (SH3) domain of the PACSIN C-terminus [77]. PACSIN 3 inhibits TRPV4 activity and affects its modulation by cell swelling and heat, whereas activation of the channel by exogenous ligands seems unaffected [77]. Introduction of a single mutation in PACSIN 3 (P142A or P143L) or mutation of specific proline residues in the proline-rich domain of TRPV4 rendered the channel insensitive to PACSIN 3-mediated inhibition [77].

Surface expression of TRPV4 is regulated by ubiquitination, mediated by a HECT-family ubiquitin ligase, AIP4 [105], which ubiquitinates TRPV4 within the stretch residing between amino acid residues 411 to 437 in the N-terminal cytoplasmic domain [105]. This ubiquitination does not target the channel for degradation, but affects the endocytic trafficking of TRPV4 (possibly via interaction with β-Arrestin [106]), and thus increases the intracellular pool of the channel [105].

### 3.2. Phosphorylation of TRPV4

A range of different kinase activities have been proposed to affect TRPV4 activity, or—more specifically—to modulate the channel activity. Activation of protein kinases A (PKA), C (PKC) and Src family kinases leads to enhanced TRPV4 activation by cell swelling in smooth muscle cells and by exposure to arachidonic acid in endothelial cells [107]. These reports point to a direct activation of TRPV4 by phosphorylation of specific amino acids on the N-terminus (Ser162, Thr175, Ser189 by PKC and Ser184 by PKA) or the C-terminus (Ser824 by PKA) [108,109]. This C-terminal phosphorylation site is specific to TRPV4 as it is an evolutionarily conserved residue that cannot be aligned to any threonine or serine in other members of the TRPV channel family. Kinase activation may lead to direct phosphorylation of a given protein or this may occur indirectly via phosphorylation of intermediate actors. While the basal activity of TRPV4, when expressed in isolation in *Xenopus laevis* oocytes, was enhanced upon PKC activation (and not PKA and PKG), none of these kinases enhanced the volume-sensitivity of the channel [63]. It is therefore likely that TRPV4 phosphorylation, in itself, does not directly modulate or activate TRPV4 channel activity, but—rather—that kinase activity affects other modulatory intermediates in a given cell type, which indirectly may affect TRPV4 function.

## 4. (Patho)Physiological Function

TRPV4, with its ubiquitous expression, has an orchestra of functions in normal physiology ascribed to it. In general, the channel is believed to play an essential role in regulating cell function by mediating Ca^2+^ influx [2,110], which enables regulation of a multitude of intracellular proteins that are required for supporting diverse physiological processes.

TRPV4-deficient mice have been studied to understand the various roles of TRPV4 in physiology [111]. Liedtke and colleagues created the first trpv4^–^/^–^ mouse by flanking exon 12 with a neo-cassette and loxP sites and excising exon 12, rendering the TRPV4 polypeptide non-functional and targeted for degradation [24]. A second trpv4^–^/^–^ mouse was generated in the laboratory of Suzuki et al. [112], in which exon four was excised using a neo-cassette marker resulting in a lack of TRPV4 production. These trpv4^–^/^–^ mice are viable and fertile up to 1 year of age and present with wide-ranging phenotypic features well-aligned with its near-ubiquitous expression in the mammalian body. They display a larger bladder capacity due to an impaired stretch and pressure sensing in the bladder wall [21,112], an overall inability to thermoregulate in stressed conditions [113,114], altered osmosensation with a subsequent reduction in water intake [24,115], compromised pain sensing [116], hearing deficiency [117], and compromised vasodilation [39]. Additionally, the trpv4^–^/^–^ mice exhibit thicker bones due to impaired osteoclast differentiation [118]. The late stage of osteoclast differentiation critically depends on TRPV4-mediated Ca^2+^ signaling, which regulates the Ca^2+^/calmodulin signaling required for sufficient osteoclast function [119,120].

### 4.1. TRPV4 in Pathophysiology

While the TRPV4 knockout mice present with relatively mild phenotypes [21,24,117,119], the TRP channels have emerged as novel and untapped pharmacological targets for a variety of diseases in several physiologic systems of the body [121,122]. As of today, we still do not know the exact role of TRPV4 in pathophysiology, although a wealth of human disease-causing mutations have been detected in the *trpv4* gene. The majority of the mutations do not map to a single region, but are distributed throughout the coding regions of the *trpv4* gene (Figure 3). The clinical manifestations of a large number of TRPV4 mutations in human patients have been described—a number that continues to grow.

#### 4.1.1. Channelopathies

Mutations in TRPV4 lead to a variety of conditions including different types of skeletal dysplasia (affected bone growth), brachydactyly (shortness of fingers and toes), familial digital arthropathy brachydactyly (FDAB), a progressive osteoarthropathy [123,124], as well as neuropathies including sensory and motor defects [125]. TRPV4-related diseases generally demonstrate autosomal dominant inheritance, but some disease-causing mutations have incomplete and variable disease penetration [126]. Curiously, and in line with the mild phenotype of the trpv4^-/-^ mice, the majority of the disease-causing mutant TRPV4 ion channels are reported as gain-of-function mutations, which lead to higher basal and evoked whole-cell Ca^2+^ currents in heterologous expression systems [6,127]. While some animal models (mice and zebrafish) expressing these gain-of-function mutations (Table 1) have demonstrated skeletal dysplasia-like phenotypes somewhat similar to the human diseases [102,119], these models still do not fully recapitulate the variety of TRPV4-induced disease presentation observed in human patients.

#### 4.1.2. Skeletal Dysplasias

One subset of TRPV4 mutations causes different variants of skeletal dysplasias, which encompass a diverse group of more than 200 diseases. Common for these conditions is defective bone and cartilage growth with patients presenting with short stature, vertebral platyspondyly (flattening of the vertebrae), scoliosis, and defects in bone ossification as the most common symptoms. *Autosomal dominant brachyolmia type 3* arises with mutations in the TM4–TM5 interconnecting loop (see Table 1 and Figure 3 for an overview of disease-causing mutations) [128]. Two such mutations (R616Q and/or V620I) lead to elevated basal TRPV4 activity with maintained mechanosensitivity [124,127,129]. *Spondylo-epimetaphyseal dysplasia Maroteaux pseudo-Morquio type 2* (SEDM-PM2) arises with mutations in the C- and N-termini and TM5 (Table 1), where the mutation E797K causes constitutively active TRPV4 channels [130]. *Spondylometaphyseal dysplasia Kozlowski* (SMDK) occurs with mutations throughout the gene (Table 1), while *Parastremmatic dwarfism* is caused by mutations in the interconnecting TM4–TM5 loop and the C-terminus (Table 1), which lead to a deficit in bone mineralization. *Metatropic dysplasia*, caused by mutations spread across the *trpv4* gene with clusters in the N- and C-termini (Table 1), is a skeletal phenotype presenting with various other complications including respiratory defects. *Familial digital arthropathy brachydactyly* (FDAB) is a relatively mild skeletal disease with no dysplasia symptoms, but with progressive osteoarthropathy (clubbing and thickening of the small hand joints). This disease is linked to TRPV4 mutations located in one of the ankyrin repeat domains (Table 1), where they cause reduced TRPV4 channel activity, and thus are loss-of-function mutations [124].

#### 4.1.3. Neuropathies

Some TRPV4 mutations cause degeneration of motor and sensory neurons [125,139], and generally encompass disturbances in distal limb function, vocal cord paresis, hearing defects and/or bladder hyperactivity [125,139,140,141,142,143]. *Congenital distal spinal motor neuropathy* (CDSMA) is caused by N-terminal mutations (see Table 2 and Figure 3 for an overview of disease-causing mutations), and it is characterized by a motor neuron deficiency leading to muscle atrophy in the lower part of the body [126]. *Scapuloperoneal spinal muscular atrophy* (SPSMA) is predominantly caused by N-terminal mutations (Table 2) [144]. Besides the general features of TRPV4 neuropathies, this condition is characterized by a reduction in shoulder blade muscles causing the typical appearance of “scapular winging” [145]. *Charcot–Marie–Tooth disease type 2C* (CMT2C) is an axonal form of Charcot–Marie–Tooth disease [140,141,142,146], originating from TRPV4 mutations in different regions of the TRPV4 channel (Table 2) [136,147,148].

### 4.2. TRPV4 Mutations and Their Genotype–Phenotype Association

Generally, there are no fundamental differences in the positions and/or patterns of amino acid substitutions within the two disease spectrums [4]. It is noticeable that TRPV4 channelopathies display a striking phenotypic variability, despite the disease-causing mutations being located in the same channel domains. While this variability is readily observed within the groups of mutations causing neuropathies and skeletal dysplasias, a few specific mutations (localized to N and C termini and the TM5) can give rise to phenotypes falling within either of the disease categories—or a combination thereof [131]:

A217S:*Spondylometaphyseal dysplasia Kozlowski* and *Scapuloperoneal spinal muscular atrophy*.

E278K:*Spondylometaphyseal dysplasia Kozlowski*/*Metatropic* dysplasia and *Scapuloperoneal spinal muscular atrophy*.

V620I:*Autosomal dominant brachyolmia type 3*and*Scapuloperoneal spinal muscular atrophy*.


P799R:
*Spondylo-epimetaphyseal dysplasia Maroteaux pseudo-Morquio type/Parastremmatic dwarfism and Charcot–Marie–Tooth disease type 2C.*


This overlapping genotype–phenotype relation suggests that the underlying pathogenic mechanisms of skeletal and nerve TRPV4 channelopathies are not always mutually exclusive.

The relatively mild phenotype of the trpv4*^-/-^* mice and the lack of obvious undesirable side-effects of systemically delivered TRPV4 inhibitors to mice and rats [101] suggest that gain-of-function mutations in TRPV4 underlie the majority of the disabling, or even lethal, human diseases. Whether it is a matter of compensatory mechanisms occurring with potential loss-of-function mutations or solely gain-of-function mutations causing disease, the severe pathologies observed with mutations in the *trpv4* gene underscore the vital role that TRPV4 function plays in regulation of diverse cellular processes.

## 5. Conclusions

Although it is well established that mutations in the *trpv4* gene are disease-causing, we have little understanding of the exact pathophysiological mechanisms underlying these diseases. With the multitude of manners by which TRPV4 can be activated, along with its many possible interaction partners, it remains unresolved which cellular signaling pathways are malfunctioning in the diseased individuals: how does altered TRPV4 function lead to the different disease symptoms? Which other cellular factors influence disease manifestations? The development of targeted and effective therapy towards hyperactive TRPV4 channelopathies depends, in part, on a more in-depth understanding of the aetiology of these diseases, the identification of the functional importance of this ion channel, and the differential regulation of tissue-specific TRPV4 function.

## Figures and Tables

**Figure 1 cells-10-00165-f001:**
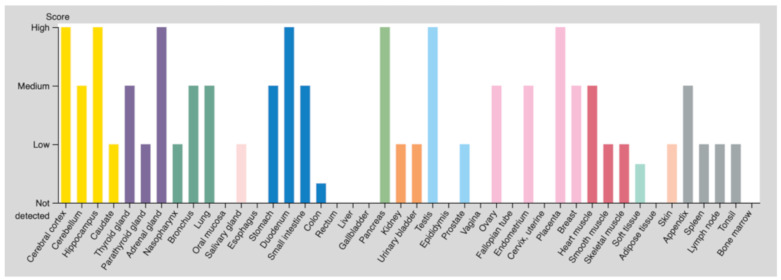
Transient receptor potential vanilloid 4 (TRPV4) (protein) expression overview. Color-coding is based on tissue groups, each consisting of tissues with functional features in common. From the Human Protein Atlas (www.proteinatlas.org).

**Figure 2 cells-10-00165-f002:**
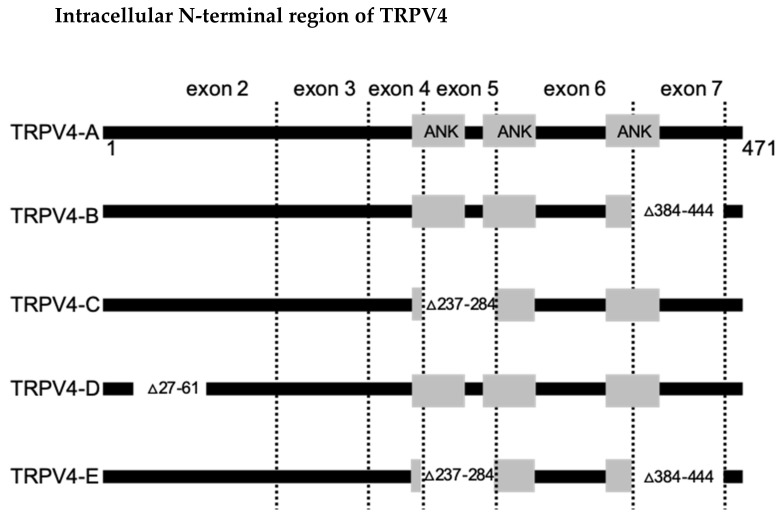
Identifications of (human) TRPV4 splice variants. Intracellular N-terminal region of the human TRPV4 channel (amino acids 1–471). Amino acids and exons lost in each variant are indicated by numbers. From [45] with permission.

**Figure 3 cells-10-00165-f003:**
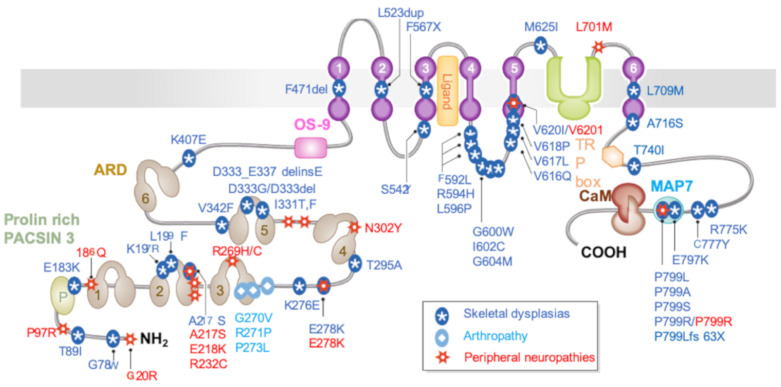
Structure of TRPV4, with putative functional domains and disease-causing mutations. From (6) with permission.

**Table 1 cells-10-00165-t001:** Mutations causing different variants of skeletal dysplasias.

Channelopathy	Mutations	References
*Autosomal dominant brachyolmia type 3*	S542Y, Y602C, R616Q, V620I, L709M	[124,127,128,129]
*Spondylo-epimetaphyseal dysplasia Maroteaux pseudo-Morquio type*	E183K, Y602C, E797K, P799R, P799L/P799Lfs63X	[123,130,131,132,133,134,135]
*Spondylometaphyseal dysplasia Kozlowski*	A217S, E278K, I331T, D333G, L523dup, R594H, L596P, G600W, F617L, M625I, L709M, A716S, C777Y, E797K	[47,130,131,132,133,136,137]
*Parastremmatic dwarfism*	R594H, E797K, P799R	[131,132,133,135,137]
*Metatropic dysplasia*	G78W, T89I, K197R, L199F, K276E, E278K, T295A, I331T, I331F, D333-E337delinsE, V342F, K407E, K471del, F592L, I694M, F617L, L618Q, T740I, R775K, E797K, P799S, P799A, P799L	[47,123,131,132,133,134,135,136,137,138,139]
*Familial digital arthropathy brachydactyly*	G270V, R217P, F272L	[124,136]

**Table 2 cells-10-00165-t002:** Mutations causing different types of neuropathies.

Neuropathy	Mutations	References
*Congenital distal spinal motor neuropathy*	G20R, P97R, R232C, R269C, R269H, R315W, R316C	[123,136,142,145,149,150]
*Scapuloperoneal spinal muscular atrophy*	A217S, R232C, R269C, R269H, E278K, R315W, R316C, V620I	[47,124,125,131,141,142,144,145,147,148]
*Charcot–Marie–Tooth disease type 2C*	R186Q, E218K, R232H, R232C, R269C, R269H, N302Y, R315W, R316C, R316H, Y567X, T701I, P799R, P799Lfs63X	[124,131,135,136,137,140,142,144,145,147,148,151]

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
