# Peer review of "TRPing to the Point of Clarity: Understanding the Function of the Complex TRPV4 Ion Channel"

_cells, 2021, doi:10.3390/cells10010165_

Round 1

Reviewer 1 Report

The review is quite informative, reflects the main points regarding the TRPV4 ion channel. However, it would be more convenient for me to work with text if the sections were numbered, and not highlighted in different fonts. I recommend that authors structure their work.

It confuses me that the main literary sources refer to the years before 2011. There is more new information and I would like it to be reflected in the review. By no means do I insist that these links be added, these are just a few examples. I recommend looking at more recent literary sources.

Peng S, Grace MS, Gondin AB, Retamal JS, Dill L, Darby W, Bunnett NW, Abogadie FC, Carbone SE, Tigani T, Davis TP, Poole DP, Veldhuis NA, McIntyre P. The transient receptor potential vanilloid 4 (TRPV4) ion channel mediates protease activated receptor 1 (PAR1)-induced vascular hyperpermeability. Lab Invest. 2020 Aug;100(8):1057-1067. doi: 10.1038/s41374-020-0430-7.

Kittaka H, Yamanoi Y, Tominaga M. Transient receptor potential vanilloid 4 (TRPV4) channel as a target of crotamiton and its bimodal effects. Pflugers Arch. 2017 Oct;469(10):1313-1323. doi: 10.1007/s00424-017-1998-7. 

Tamara Rosenbaum, Miguel Benítez-Angeles, Raúl Sánchez-Hernández, Sara Luz Morales-Lázaro, Marcia Hiriart, Luis Eduardo Morales-Buenrostro and Francisco Torres-Quiroz. TRPV4: A Physio and Pathophysiologically Significant Ion Channel. Int. J. Mol. Sci. 2020, 21, 3837; doi:10.3390/ijms21113837

Author Response

Reviewer #1

The review is quite informative, reflects the main points regarding the TRPV4 ion channel. However, it would be more convenient for me to work with text if the sections were numbered, and not highlighted in different fonts. I recommend that authors structure their work.

It confuses me that the main literary sources refer to the years before 2011. There is more new information and I would like it to be reflected in the review. By no means do I insist that these links be added, these are just a few examples. I recommend looking at more recent literary sources.

Peng S, Grace MS, Gondin AB, Retamal JS, Dill L, Darby W, Bunnett NW, Abogadie FC, Carbone SE, Tigani T, Davis TP, Poole DP, Veldhuis NA, McIntyre P. The transient receptor potential vanilloid 4 (TRPV4) ion channel mediates protease activated receptor 1 (PAR1)-induced vascular hyperpermeability. Lab Invest. 2020 Aug;100(8):1057-1067. doi: 10.1038/s41374-020-0430-7.

Kittaka H, Yamanoi Y, Tominaga M. Transient receptor potential vanilloid 4 (TRPV4) channel as a target of crotamiton and its bimodal effects. Pflugers Arch. 2017 Oct;469(10):1313-1323. doi: 10.1007/s00424-017-1998-7. 

Tamara Rosenbaum, Miguel Benítez-Angeles, Raúl Sánchez-Hernández, Sara Luz Morales-Lázaro, Marcia Hiriart, Luis Eduardo Morales-Buenrostro and Francisco Torres-Quiroz. TRPV4: A Physio and Pathophysiologically Significant Ion Channel. Int. J. Mol. Sci. 2020, 21, 3837; doi:10.3390/ijms21113837

Answer: We thank the reviewer for the positive comments. We have structured the text differently with each section numbered as requested by the reviewer and cited more recent literature.

Reviewer 2 Report

This is an interesting and comprehensive review of knowledge about TRPV4 isoforms and channels. It covers all the major facets of TRPV4 physiology and biophysics and, as such, will serve as a useful reference for others interested in TRPV4 biology. My only minor concern is that the language usage in the latter part of the paper has several errors (lines 155,234,258,263 and 269-270). The last lines mentioned are difficult enough to understand that the meaning of the sentence is unclear. Otherwise the manuscript is acceptable as is.

Author Response

Reviewer #2

This is an interesting and comprehensive review of knowledge about TRPV4 isoforms and channels. It covers all the major facets of TRPV4 physiology and biophysics and, as such, will serve as a useful reference for others interested in TRPV4 biology. My only minor concern is that the language usage in the latter part of the paper has several errors (lines 155,234,258,263 and 269-270). The last lines mentioned are difficult enough to understand that the meaning of the sentence is unclear. Otherwise the manuscript is acceptable as is.

Answer: We thank the reviewer for the positive comments. We have, as requested by the reviewer, revised the parts of the text mentioned here (marked in red in the revised version).

Reviewer 3 Report

Summary:

This is a very well written and essential review regarding the multifaceted roles of TRPV4 in mediating control of tissue function in health and disease. In addition, the review provides a very complete description of relationships between structural TRPV4 disruptions resulting from altered gene expression that generate different pathophysiological consequences compromising tissue homeostasis. It will serve as an excellent reference source for investigators in different fields of interest focused on clarifying signaling mechanisms mediating TRPV4 control of tissue function in health and disease.  One example of its contribution is that it will be undoubtedly instrumental in guiding future efforts to better understand why gain of TRPV4 function may have more severe effects on tissue homeostasis than loss of its function. There are only two minor changes which will be helpful in improving the completeness of its coverage of the literature dealing with TRPV4 involvement in controlling numerous responses in health and disease.

Minor Concerns:

  1. Literature Inclusions: There is a large body of literature regarding TRPV4 involvement in controlling ocular tissues, This review needs to cite at the very least references regarding the importance of TRPV4 expression in other ocular tissues besides just the retina. Pub Med lists 66 publications dealing with TRPV4 involvement in the eye. As a starting point, they will find it very helpful to cite an excellent new review just published this month in Experimental Eye Research. The role of TRPV4 channels in ocular function and pathologies. Guarino BD, Paruchuri S, Thodeti CK. PMID 32979394

  1. Grammatical errors: a) Line 302 change defect to defective; b) Line 330 and it (add) is characterized; c) Line 336 originating from trpv4 (not by); d) Line 348: can give rise to (add)

Author Response

Reviewer #3

This is a very well written and essential review regarding the multifaceted roles of TRPV4 in mediating control of tissue function in health and disease. In addition, the review provides a very complete description of relationships between structural TRPV4 disruptions resulting from altered gene expression that generate different pathophysiological consequences compromising tissue homeostasis. It will serve as an excellent reference source for investigators in different fields of interest focused on clarifying signaling mechanisms mediating TRPV4 control of tissue function in health and disease.  One example of its contribution is that it will be undoubtedly instrumental in guiding future efforts to better understand why gain of TRPV4 function may have more severe effects on tissue homeostasis than loss of its function. There are only two minor changes which will be helpful in improving the completeness of its coverage of the literature dealing with TRPV4 involvement in controlling numerous responses in health and disease.

Minor Concerns:

  1. Literature Inclusions: There is a large body of literature regarding TRPV4 involvement in controlling ocular tissues, This review needs to cite at the very least references regarding the importance of TRPV4 expression in other ocular tissues besides just the retina. Pub Med lists 66 publications dealing with TRPV4 involvement in the eye. As a starting point, they will find it very helpful to cite an excellent new review just published this month in Experimental Eye Research. The role of TRPV4 channels in ocular function and pathologies. Guarino BD, Paruchuri S, Thodeti CK. PMID 32979394

  1. Grammatical errors: a) Line 302 change defect to defective; b) Line 330 and it (add) is characterized; c) Line 336 originating from trpv4 (not by); d) Line 348: can give rise to (add)

Answer: We thank the reviewer for the positive comments. We have, as requested by the reviewer, corrected the grammatical errors (marked in red in the revised version), and cited more literature concerning the role of TRPV4 in ocular tissues.

Round 2

Reviewer 1 Report

The text of the article is really structured, the sections are numbered, which makes it easier to work with the article. However, with regard to links to recent publications, only 2 works have been added to the list of references, which, in my opinion, is not an update. I believe that a review with links from ten years ago cannot pretend to be relevant and new.

Author Response

Answer: We thank the reviewer for valuable comments and agree with the reviewer. Recent literature is of high importance, and we have included more publications. However, it is by no means our intention to neglect or exclude recent literature, but several articles concerning TRPV4 in diabetes, cancer or e.g. edema are not included as our review does not go in depth with these pathological conditions.

Currently we have 142 references.

From 2010 and earlier: 73 references (51 % of the total number of references).

From 2011 – today: 69 references (49 % of the total number of references).

The more recent (from 2011 and onwards) publications are distributed as following:

2011 – 2015: 59 %

2016 – 2021: 41 %

We hope to have met the critique point from the reviewer. If we are still missing some obvious publications we kindly ask the reviewer to guide us towards them.